# Novel symmetrical bifacial flexible CZTSSe thin film solar cells for indoor photovoltaic applications

Hui Deng [1], Quanzhen Sun[1], Zhiyuan Yang[1], Wangyang Li[1], Qiong Yan[1], Caixia Zhang[1,2], Qiao Zheng[1,2], Xinghui Wang[1], Yunfeng Lai[1,2] & Shuying Cheng [1,2 ✉]

Environment-friendly flexible $Cu_2ZnSn(S,Se)_4$ (CZTSSe) solar cells show great potentials for indoor photovoltaic market. Indoor lighting is weak and multi-directional, thus the researches of photovoltaic device structures, techniques and performances face new challenges. Here, we design symmetrical bifacial CZTSSe solar cells on flexible Mo-foil substrate to efficiently harvest the indoor energy. Such devices are fabricated by double-sided deposition techniques to ensure bifacial consistency and save cost. We report 9.3% and 9% efficiencies for the front and back sides of the flexible CZTSSe solar cell under the standard sun light. Considering the indoor environment, we verify weak-light response performance of the devices under LED illumination and flexibility properties after thousands of bending. Bifacial CZTSSe solar cells in parallel achieve the superposition of double-sided output current from multi-directional light, significantly enhancing the area utilization rate. The present results and methods are expected to expand indoor photovoltaic applications.

[1] College of Physics and Information Engineering, Institute of Micro-Nano Devices and Solar Cells, Fuzhou University, Fuzhou, China. [2] Jiangsu Collaborative Innovation Center of Photovoltaic Science and Engineering, Changzhou, China. ✉email: sycheng@fzu.edu.cn

Kesterite Cu$_2$ZnSn(S,Se)$_4$ (CZTSSe) with superior optoelectronic properties, tunable band gaps, and earth-abundant elementals has attracted extensive attentions for thin film solar cells[1–6]. Flexible CZTSSe solar cells due to their non-toxicity, lightweight and high stability show great potentials in applications such as portable equipment, military fields and building-integrated photovoltaic[7,8]. In recent years, flexible CZTSSe thin film solar cells have obtained great progresses in the aspects of power conversion efficiency (PCE) and substrate optimization. The flexibility and performance are strongly dependent on substrates including flexible glass[9], stainless steel[10,11], and Mo foil[12,13]. The PCEs of flexible CZTSSe solar cells on Mo foil and stainless steel substrates by sputtering deposition both reached 10.3%[10,14]. Our group focuses on the study of Mo-foil-based flexible CZTSSe solar cells and has obtained 7.19% efficiency by technique optimization[15–17]. Mo foils with high conductivity and bendability can be directly used as electrodes and flexible supported substrates, reducing additional process costs. It is of great significance to investigate CZTSSe solar cells based on Mo foils for flexible energy devices with low cost and high efficiency.

Indoor photovoltaic applications with flexible devices gradually come into sight for developing smart houses and industries by absorbing lights and saving energy[18,19]. The indoor lights are usually weak in all directions, which are difficult to be effectively utilized. The reported substrate-structured Mo-based CZTSSe solar cells exhibited good performance under low-light conditions[20] but limited by light directions. Bifacial solar cells have unique advantages in collecting front and rear illuminations, reducing the solar cell cost in a photovoltaic system by efficiently utilizing materials and areas[21–23]. For traditional bifacial solar cells, the absorption layer was fabricated on the transparent conductive substrate such as ITO, FTO, and nano-metals[24–26]. The bifacial CZTSSe solar cells with the structure of glass/FTO/CZTSSe/CdS/i-ZnO/ITO obtained 6.3% front efficiency and 1.1% rear efficiency[27], showing large performance differences in double-sided illuminations. The sacrifice of Mo layer is not beneficial to the performance of CZTSSe solar cells[28]. Meanwhile, FTO and ITO glasses are easily affected by high-temperature CZTSSe process, which is hard to be compatible with flexible systems. In contrast, flexible Mo foil substrates as conductors have enough hardness to support the deposition of double-sided layers. Symmetrical bifacial flexible CZTSSe thin film solar cells on Mo foils can receive indoor multi-directional illumination and reduce space occupation. And the devices can be designed into different shapes for indoor ornament integrated photovoltaics.

Here, we present a novel symmetrical bifacial flexible CZTSSe solar cells with high performance and bendability for indoor photovoltaic applications. The front-sided and back-sided solar cells are symmetrically deposited on a Mo foil using simultaneous one-time process. The bifacial solar cell with the same structure and morphology obtain the PCEs of 9.3% and 9% for two sides, respectively. Under weak LED light illuminations with 1.5–18.5 mW/cm$^2$ intensities, the devices also show good performance of 6–8.8% efficiency. The double sides respectively bearing stretching and squashing stresses are used to study the effects of bottom peeling off and grain boundary cracks on device performance. The flexible CZTSSe devices can stand large-angle and thousands of bending at stretching and squashing states without noticeable PCE degradation. For parallel circuit of bifacial solar cells, the output current (8.7 mA) achieved obvious superposition of front (7 mA) and back cells (2.1 mA) at different light intensities. And the parallel devices directly produce a periodically variable current by substrate rotation at all-direction illuminations, enhancing the area utilization rate.

## Results

**Design of bifacial flexible CZTSSe solar cells**. Bifacial thin film solar cells are not limited by illumination directions, showing great potentials in narrow environment and indoor photovoltaics. The bifacial solar cell structure can be designed as overhung devices as shown in Fig. 1a. As the sun rises in the east and sets in the west, both two sides of solar cells can absorb sunlight and transfer it to photocurrent for whole days. The overhung bifacial solar cells assisted with external circuits can effectively save occupied area and improve sunlight utilization. The structures reveal large advantages for photovoltaic applications in some specific environments. On the other hand, bifacial devices based on flexible substrate can be made into different shapes for indoor ornament integrated photovoltaics as shown in Fig. 1b. The devices effectively collect and utilize the weak and divergent lights for convenient life. Considering the motivation of environmental friendliness and realizability, we design a new double-sided structure and study the performance of bifacial flexible CZTSSe solar cells. The devices with double-sided symmetrical structure (Ag/ITO/ZnO/CdS/CZTSSe/MoSe$_2$/Mo foil/MoSe$_2$/CZTSSe/CdS/ZnO/ITO/Ag) are shown in Fig. 1c. The Mo foil is the preferred flexible substrate matching well with CZTSSe solar cells because of superb double-sided conductivity and bendability. The substrate-structured CZTSSe solar cells at two sides are connected by one common Mo foil substrate. The front and back side can

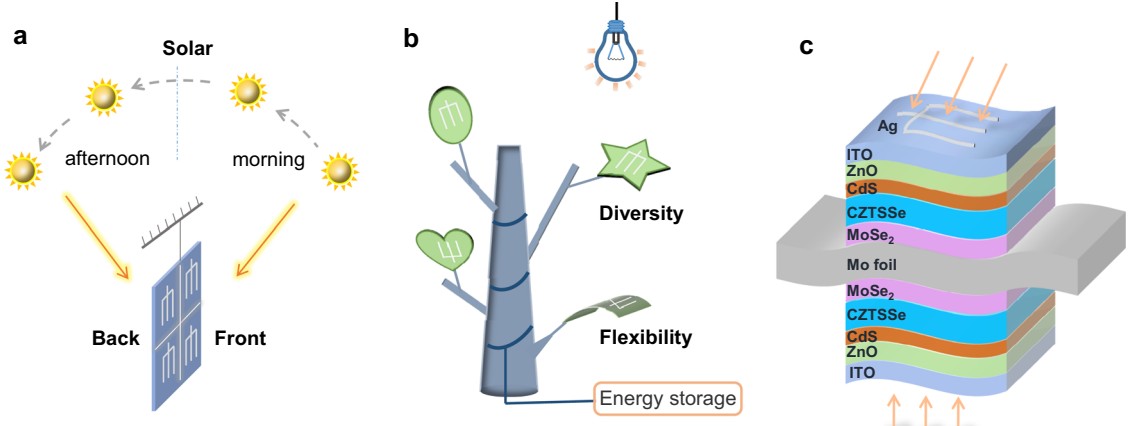

**Fig. 1 Applications and structure of bifacial solar cells. a** The diagram of overhung bifacial solar cells. **b** The diagram of indoor photovoltaic application using flexible bifacial solar cells. **c** Device structure diagram of bifacial flexible CZTSSe solar cells.

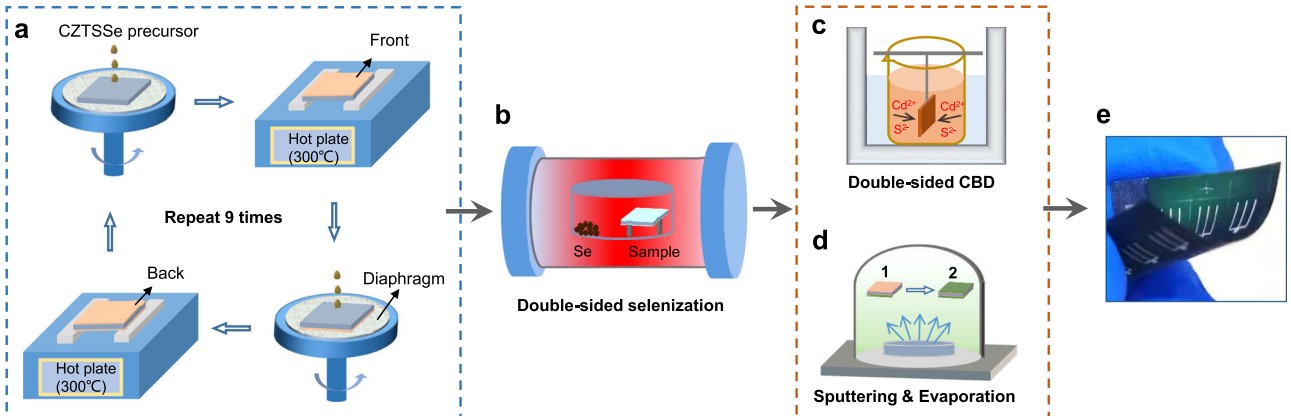

**Fig. 2 Schematic illustration of the double-sided deposition techniques. a** Circularly spin-coating precursor solution and annealing for deposition of double-sided CZTSSe prefabricated film. **b** Selenization treatment for double-sided crystalline CZTSSe film. **c** The CBD process for deposition of double-sided CdS film. **d** Vacuum methods including sputtering and evaporation for fabrication of double-sided ZnO/ITO/Ag layers. **e** The photograph of bifacial flexible CZTSSe solar cell.

independently work at opposite illumination and jointly output current by parallel circuit.

**Double-sided deposition techniques.** To ensure bifacial consistency and decrease costs, the same layers on two sides of a Mo foil are deposited as the same process at one-time operation. The fabrication techniques of bifacial CZTSSe solar cells are shown in Fig. 2 as detailedly described in the "Methods" section. The double sides of substrates are simultaneously cleaned by electrochemical etching. Then the double-sided CZTSSe layers are deposited by cyclic spin-coating method (see Fig. 2a). The side of Mo foil first spin-coated precursor solution is marked as the front side and the other is back side. In order to keep the samples clean, we added fresh reticular diaphragms for each spin-coating and supporting frames for annealing. The spin-coating and annealing process are repeated 9 times by alternating on front and back sides to obtain double-sided prefabricated CZTSSe films. Next, selenization treatment of prefabricated film is realized in a $N_2$ filled furnace with Se powder at 550 °C. As shown in Fig. 2b, the sample is set on stilts and exposed the front and back surfaces in Se atmosphere to ensure the same selenization process for two sides. The double-sided $MoSe_2$ layers and crystalline CZTSSe films (Se rich) can be obtained from the selenization process. As shown in Fig. 2c, the double-sided CdS buffer layers are deposited by chemical bath deposition (CBD) method. In solution, $Cd^{2+}$ and $S^{2-}$ can adsorb simultaneously on the surface of front and back CZTSSe films to form uniform CdS layers for one experiment. Finally, ZnO/ITO/Ag layers are fabricated by vacuum methods including sputtering and evaporation (see Fig. 2d), which need to be completed with one side followed by the other side. The photograph of obtained bifacial flexible device is shown in Fig. 2e, indicating that the devices are uniformly distributed on the two sides with good flexibility. There are 9 independent devices for each side with active area of 0.21 cm². It is found that the double-sided fabrication techniques including substrate cleaning, spin-coating, annealing, selenization, and chemical bath deposition are respectively operated on one-time experiments for two sides, saving time and avoiding accidental deposition differences.

**Characterizations of bifacial flexible CZTSSe solar cells.** The scanning electron microscope (SEM) measurements with energy dispersive spectroscopy (EDS) are conducted to investigate the morphology and structure characteristics of the bifacial devices. The image of cross section prepared by ion beam cutting is shown

in Fig. 3a. The front and back multiple layers are evenly distributed on both sides of Mo-foil substrate with the whole thickness of 62 μm. The front cell almost same with the back one manifests uniform morphology and thickness (~5 μm). The CZTSSe films after selenization on surface of both two sides are smooth and compact with clear grains (see Fig. 3b and d). The surface large-grain layer of CZTSSe film with ~1 μm grains reveals high quality crystallization. The double-sided samples of CZTSSe/MoSe₂/Mo are analyzed by X-ray diffraction (XRD) spectra as shown in Fig. 3f. Except the peaks of MoSe₂ and Mo, all diffraction peaks are consistent with standard CZTSSe positions[12,29]. The obtained bifacial CZTSSe layers dominated by (112) face are pure alloy phase without any impurity or secondary phase. The magnified cross-section images from two-sided areas A1 and A2 are shown in Figs. 3c and e, respectively. Each layer of the bifacial device represented by Mo/MoSe₂/CZTSSe/CdS/ZnO/ITO is clearly visible and easily distinguishable. The surface morphologies of front and back sides are exactly the same, demonstrating reliable fabricated technique for bifacial solar cells. The thicknesses of Ag, ITO, ZnO, and CdS layers are about 400 nm, 200 nm, 60 nm, and 70 nm, respectively. The double-sided MoSe₂ layers between Mo-substrate and CZTSSe are obtained from selenization treatment whose thicknesses are about 2 μm. The CZTSSe film deposited by solution method generally shows two layers, large-grain layer and small-grain layer, consistent with the related reports[30,31]. The thicknesses of the two layers are 620 nm and 1500 nm, respectively. The EDS line scans from surface to Mo substrate of the front side are conducted to study element distributions of various layers (see Fig. 3g). The evolutions of all elements are essentially consistent with the materials of every layer. The elements of Cu, Zn, Sn, Se distributed in absorber layer can be observed. The S element signal is masked by stronger signal of Mo element. From surface to bottom of the CZTSSe film, the contents of metal cations (Cu, Zn, Sn) reduce and those of anions (Se, S) increase, which mainly cause the small-grain layer. The average composition of absorber is $Cu_2ZnSn(S_{0.17}Se_{0.83})_4$. The junction interface and back contact exist element interdiffusion, which enhances the anchoring strength with substrate. According to above characterizations, the front and back cells with complete structures form extremely same mirror distribution on a Mo foil by the stable deposition techniques.

**Photovoltaic performance of the devices.** The performances of bifacial flexible solar cells are independently measured to make comparison by detailed statistical analysis of 18 devices in each

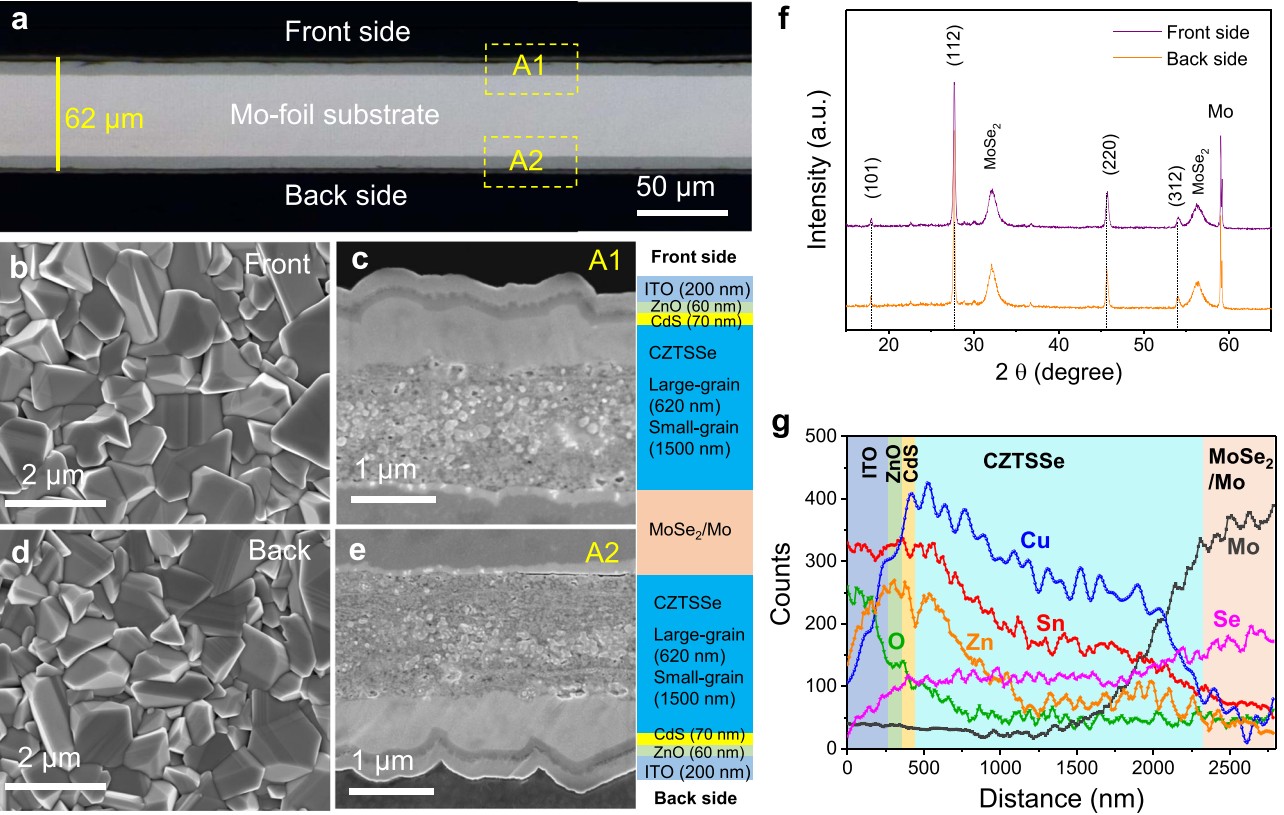

**Fig. 3 Characterizations of the bifacial flexible CZTSSe solar cells. a** Cross-section SEM image of bifacial flexible CZTSSe solar cell. Top-view SEM images of **b** front-sided and **d** back-sided CZTSSe film. The magnified cross-section images from **c** front-sided area A1 and **e** back-sided area A2. **f** XRD spectra of surfaces of double-sided CZTSSe/MoSe$_2$/Mo layers. **g** EDS line scans from device surface to Mo substrate of the front side.

group. As shown in Fig. 4a, the PCE distribution boxplots of both front and back cells locate at the region of 7.8–9.3%, indicating good uniformity of the whole bifacial solar cells. The average PCE of front cell (8.7%) is a little higher than that of the back side (8.4%). The small differences of PCEs (0.3%) between front and back cells are caused by the fill factors (FF) compared with almost same open voltage ($V_{OC}$) and short-circuit current density ($J_{SC}$) (Supplementary Fig. 1). The systemic variations caused by deposition sequence is decreased to the minimum. The $J–V$ curves and external quantum efficiency (EQE) spectra of bifacial flexible CZTSSe solar cells are shown in Fig. 4b and c, respectively. The optimal efficiency of front cell is 9.3% with a $V_{OC}$ of 436 mV, a $J_{SC}$ of 33.76 mA/cm$^2$ and an FF of 63.2%. Similarly, the back cell obtains 9% PCE with the $V_{OC}$ of 434 mV, the $J_{SC}$ of 33.7 mA/cm$^2$ and the FF of 61.7% (see Fig. 4b). Both of the double-sided devices on a Mo foil manifest excellent PCEs compared with reported flexible solar cells[13,14,32]. Meanwhile, the bifacial devices exhibit coincident EQE curves with high performances whose integral current densities reach 33.4 mA/cm$^2$ (see Fig. 4c). The EQE values are more than 80% in the wavelength range of 510–870 nm. The flexible CZTSSe solar cells can realize efficient collection and utilization for the spectra of 350–1250 nm. In addition, stability of solar cells is an important index for practical application. The light stability data of flexible CZTSSe solar cell is shown in Supplementary Fig. 2. The PCE of the solar cells obtained small increase after standing 5-min illumination and kept stable 9.3% for 1-h illumination by continuously testing. Furthermore, the unencapsulated device was put in a solar cell aging box by continuous illumination with a standard sun intensity for several days and tested once a day. After 14-day continuous illumination, the device had no obvious degradation (<5%). Flexible CZTSSe thin film solar cells using all inorganic

materials reveal high stability which is expected to realize wide application.

To check the indoor weak light response, the performances of flexible CZTSSe solar cells were tested using a LED light illumination. The $J–V$ characteristics of devices under the LED light intensities of 1.5–18.5 mW/cm$^2$ are shown in Fig. 4d. All the curves with rectification effects reveal good response to weak light. The device obtains 8.8% efficiency when the light intensity is 18.5 mW/cm$^2$. And the PCE can maintain 6% even illuminated at 1.5 mW/cm$^2$ weak LED light. The PCE, $J_{SC}$, $V_{OC}$, and FF evolutions with light intensities are shown in Fig. 4e, f and Supplementary Fig. 3. The $J_{SC}$ values show superb linear relationship with LED light intensities ($I_{LED}$) (see Fig. 4e). We fitted $J_{SC}–I_{LED}$ relationship according to the power law dependence ($J_{SC} \propto I^{\alpha}$) by logarithmic coordinates[33]. The power value $\alpha$ for the devices is 0.99 in weak light condition, close to unity (first-order). This means that the CZTSSe solar cell demonstrates good photoresponse for weak light and its loss mechanism is from trap-assisted recombination[34]. The recombination process can be reflected based on the relationship of $V_{OC} \propto n(k_{B}T/q)\ln(I)$, where $k_{B}$ is the Boltzmann constant, $T$ is the temperature and $q$ is elementary charge[35]. For trap-free solar cells, the slope of $V_{OC}$ vs. ln($I$) should be $k_{B}T/q$ (i.e., $n = 1$). The $V_{OC}$ under weak light illumination evolutions with logarithms of light intensities accord with the relationship $V_{OC} \propto 1.42(k_{B}T/q)\ln(I_{LED})$ as shown in Supplementary Fig. 3a. The larger slope indicated $V_{OC}$ would rapidly decline because of trap recombination under the extreme weak light. The decreased photogenerated carriers at weak light condition weaken the $V_{OC}$ and $J_{SC}$ of solar cells. The fill factors can keep constant at ~64% for LED illumination with 4–18.5 mW/cm$^2$ intensities (Supplementary Fig. 3b), which is mainly affected by the interface and contact of each layer. When the light

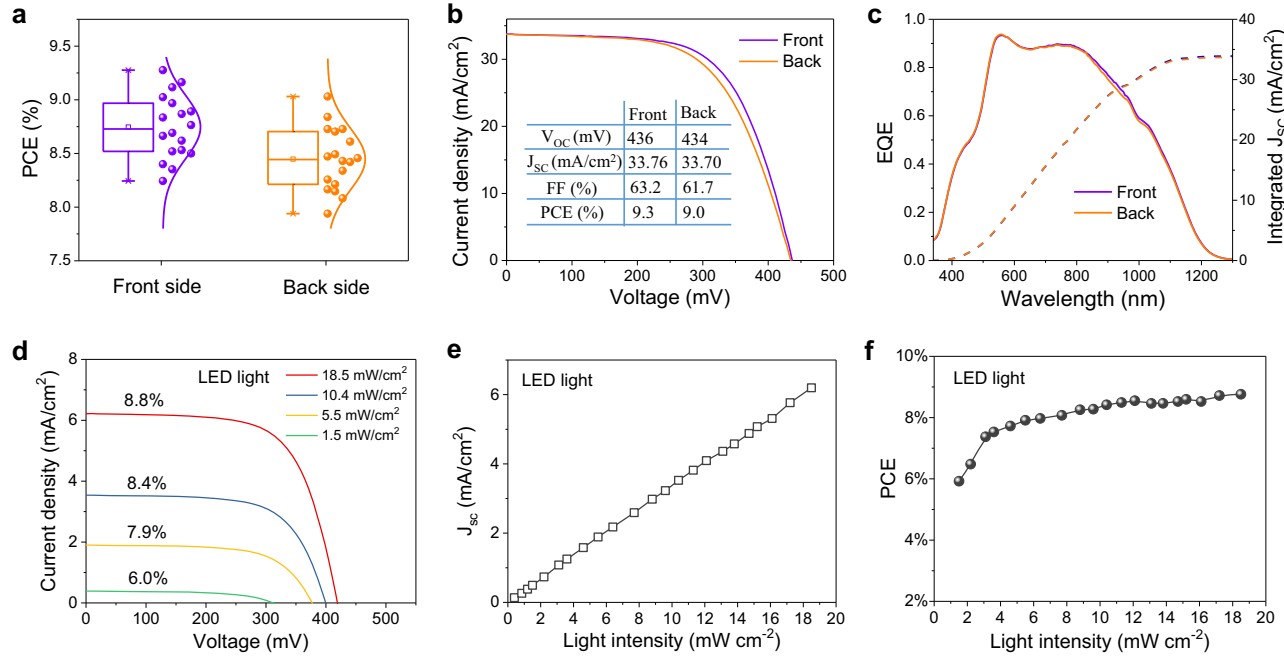

**Fig. 4 Photovoltaic device properties of bifacial flexible CZTSSe solar cells. a** Statistical PCE distribution boxplots of front and back sides on bifacial flexible CZTSSe solar cells. **b** The J–V curves and **c** EQE spectra of champion double-sided cells. **d** The J–V curves of front solar cell under the LED light illuminations with different intensities. **e** The $J_{SC}$ evolution with LED light intensity. **f** The PCE evolution with LED light intensity.

intensity is less than 4 mW/cm², the fill factors show obvious decrease. From the efficiency trend in Fig. 4f, the PCEs of flexible CZTSSe solar cells are still more than 8% under the LED light with 5–20 mW/cm² intensities. The performances manifest degradation at extreme weak light (<3 mW/cm²) mainly caused by decreased $V_{OC}$ and FF. Overall, the presented bifacial CZTSSe flexible solar cells can realize weak light response and be suitable for indoor photovoltaic application.

**Flexible properties of the bifacial solar cells**. Flexibility is important for solar cells to be applied in many occasions, particularly indoor building-integrated photovoltaics. To investigate the flexible properties, we fixed the device on a vernier caliper and controlled different bending degrees by the distance between two ends as shown in Fig. 5a. The bending degree is described as the angle of top area with horizontal line of two ends, which is estimated by the reading of vernier caliper. The bifacial flexible CZTSSe device can stand 0–70° bending angles and be restored to the original state. Once over 70°, the device deformations are unable to recover. Owing to the double-sided symmetrical structure, the front side and back side suffered different bending stresses at one synclastic operation. In Fig. 5b, we can respectively analyze the stretching and squashing performances of front and back cells by that bending state. The force points mainly fall on the grain boundaries of CZTSSe layer at stretching state of front side (see the red dashed box in Fig. 5b). While for the squashing state of the back cell, the stress point locates on the interface of substrate and CZTSSe film (see the blue dashed box in Fig. 5b). To study the effect of bending stress on device performance, the normalized PCE degradation of bifacial cells at varied bending angles are measured and shown in Fig. 5c. When the bending angles are less than 60°, both the PCEs of double sides have no obvious degradation. As the angle exceeds 60°, the PCEs of back side with large squashing stress manifest 10–20% degradation compared with the front cells. The larger-angle bending causes CZTSSe film partly detached from substrate under squashing stress (Supplementary Fig. 4b). Furthermore, the

normalized PCE degradation of the device evolved with bending numbers (0–4000 times) at a fixed angle of 50° are shown in Fig. 5d. The bifacial solar cells can keep more than 95% of original efficiencies after bending 3000 times. Before and after bending treatments, the J–V curves are almost coincident for the two sides, confirming high flexibility of the obtained solar cells (Supplementary Fig. 5). With increasing bending times, the degradation with 15% after 4000 times is from the front device, in which the crystalline CZTSSe films is gradually destroyed by stretching stress (Supplementary Fig. 4a). From the above, the flexible CZTSSe solar cells under stretching stress can stand larger-angle (>60°) bending but not endure numerous bending times (0-3000 times). On the other hand, the device under squashing stress can stand numerous bending times (>4000 times) but not endure larger-angle bending (0–60°). It is an advantage for bifacial flexible CZTSSe solar cells to study the effects on variant stress states, which provides the guide of flexible device performance optimization.

**Parallel circuit characteristics of the bifacial solar cells**. The Mo foil whole substrate as a good conductor directly connects the front and back cells, providing a parallel condition for double sides. Parallel bifacial solar cells are beneficial to improve the utilization of device area by absorbing illuminations in different directions. A simple demo shown in Fig. 6a is used to measure the output characterizations of parallel devices. Considering that average albedo of the earth is 0.3, we use a reflector to change the light direction and reflect about 0.3 times the standard sun light to the other sides. The output current–voltage (I–V) curves for the bifacial solar cells are shown in Fig. 6b. The solar cell of one side illuminated at 1 standard sun obtained 9.1% efficiency. And the other sided cell at the 0.3 times sun illumination obtained 8.7% efficiency with decreased $V_{OC}$ (see Table 1). The CZTSSe solar cell can still keep high PCE in 0.3 times the sun. The photogenerated carriers come to decrease at weak light condition, the $V_{OC}$ decrease as logarithmic trend and $J_{SC}$ decreased as linear trend with light intensity according to the former analysis. The FF

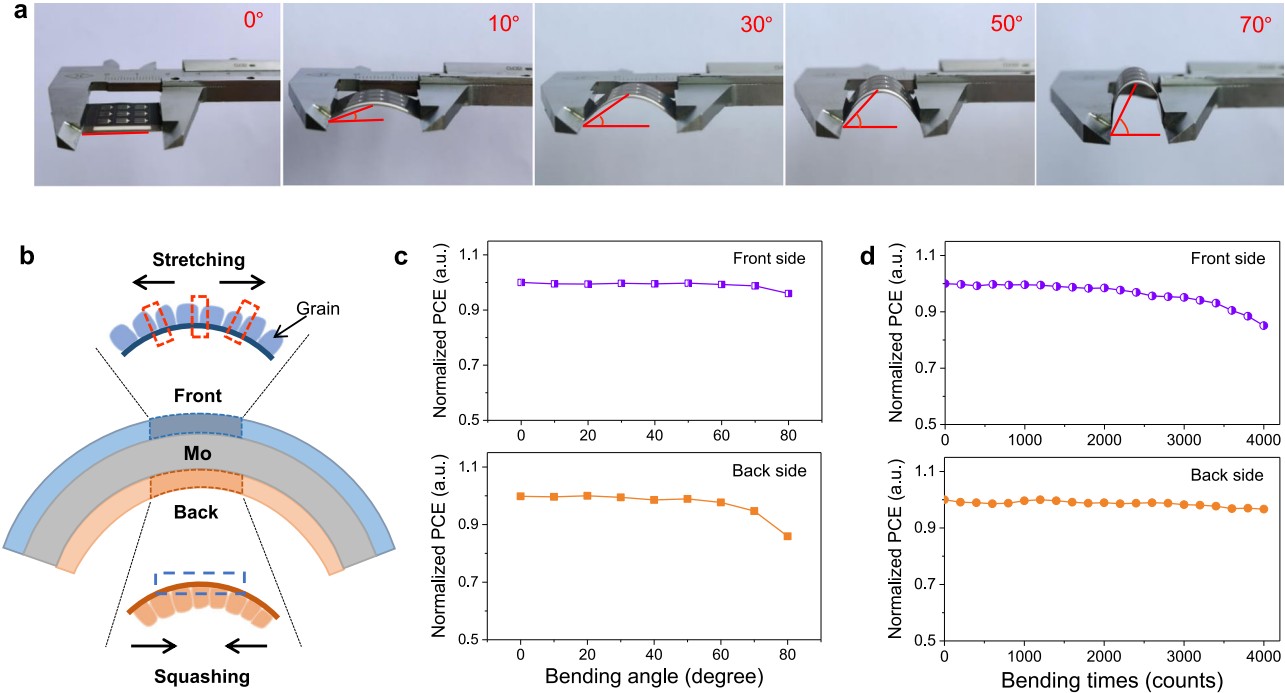

**Fig. 5 Flexible performance of bifacial CZTSSe solar cells. a** Photographs of bended bifacial CZTSSe solar cells at varied angles from 0° to 70°. **b** Schematic bending states with stretching and squashing of front and back sides. **c** Normalized PCE degradation percent of bifacial solar cells at different angles (0°–80°) with 100 time bending. **d** Normalized PCE degradation evolution with different bending cycles (0–4000 times) at a fixed angle of 50°.

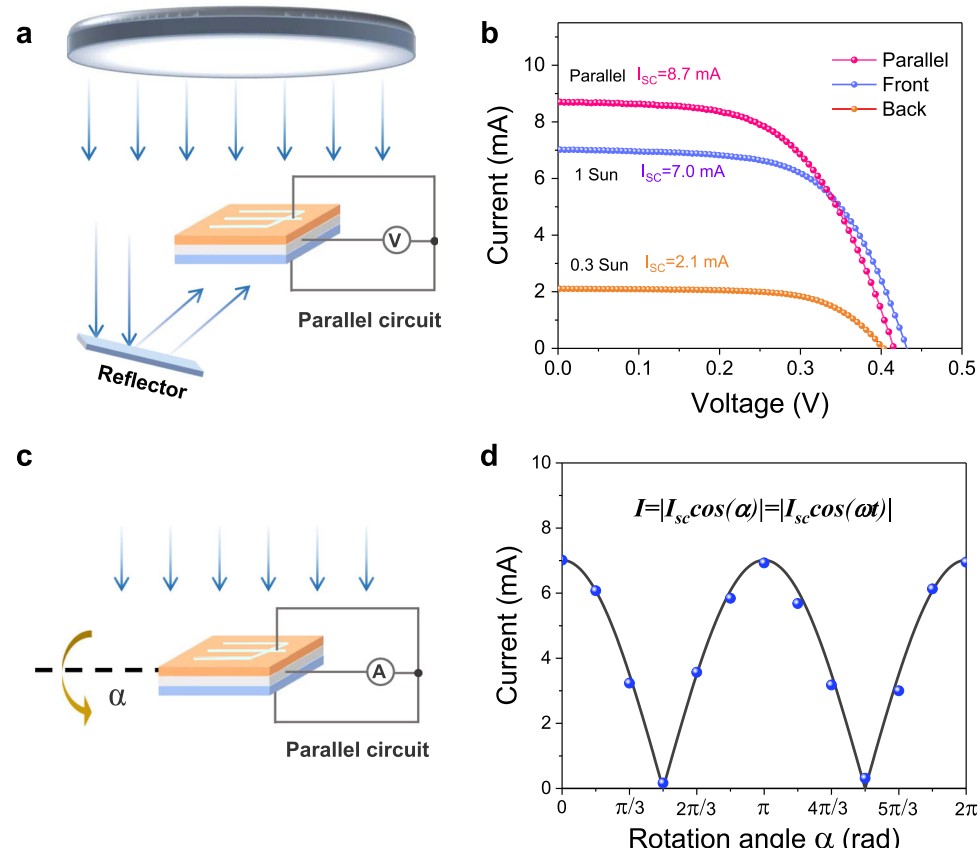

**Fig. 6 Parallel output properties of bifacial flexible CZTSSe solar cells. a** Schematic diagram of performance measurement for double sides in parallel. **b** The *I–V* curves of double-sided solar cells and parallel device. **c** Measurement schematic diagram of parallel currents at rotation angles. **d** Output parallel current evolution with rotation angles.

**Table 1 The typical photovoltaic parameters of double-sided CZTSSe solar cells in parallel circuit state.**

| Device | Lighting | $V_{OC}$ (mV) | $I_{SC}$ (mA) | $J_{SC}$ (mA/cm$^2$) | FF | PCE | $P_{max}$ (mW) |
|---|---|---|---|---|---|---|---|
| Front side | 1 sun | 432 | 7.0 | 33.4 | 63% | 9.1% | 1.87 |
| Back side | 0.3 sun | 406 | 2.1 | 10.1 | 64% | 8.7% | 0.55 |
| Parallel | / | 416 | 8.7 | / | 58% | / | 2.05 |

of the solar cell (64%) at 0.3 times sun lighting is slightly higher than that of device (63%) at one sun light. The temperature on sample at weak light condition is lower than that of 1 sun illumination, which is helpful to improve reverse saturation current for the higher fill factor. For the parallel bifacial solar cell, the $V_{OC}$ (416 mV) is between those of front (432 mV) and back cells (406 mV). The output short-circuit current ($I_{SC}$) of parallel device (8.7 mA) is roughly equal to the sum of independent currents of front cell (7.0 mA) and back cell (2.1 mA). The loss rate of output current is less than 5% in the parallel circuit. From the power curves in Supplementary Fig. 6, the maximum output power (2.05 mW) is also approximate to the sum of the two sides with about 15% loss. The front and back cells produce different $V_{OC}$ (432 mV and 406 mV, respectively) under different illuminations. In the parallel circuit, the front cell with higher $V_{OC}$ provides some voltage consumption for the back cell. Therefore, the parallel power loss is from the internal resistance consumption resulted by unequal doubled $V_{OC}$, which is acceptable for largely different intensities of two sides. The bifacial CZTSSe solar cells have revealed parallel superimposed effect for current and output power even illuminated at different illumination conditions of the two sides. Furthermore, the parallel device is used to explore the properties of output currents at continuously variable illumination directions. Under applied zero bias voltage, the output current of parallel bifacial device is tested at various rotation angles α for a circle (see Fig. 6c). The tested output current values at 30° intervals are conformed to the cosine relationship ($I = |I_{SC} \cos(\alpha)|$) as shown in Fig. 6d. The parallel bifacial CZTSSe solar cells directly converse solar energy to periodically variable currents by rotating the substrate. Therefore, the present bifacial flexible CZTSSe solar cells manifest specific indoor application potentials using weak and all-directional light.

## Discussion

In summary, the novel bifacial flexible CZTSSe solar cells are proposed to utilize sun illumination from all directions for the applications in overhung solar cells, indoor photovoltaics, outer space photovoltaics, and so on. The device with double-sided symmetrical structure (Ag/ITO/ZnO/CdS/CZTSSe/MoSe$_2$/Mo foil/MoSe$_2$/CZTSSe/CdS/ZnO/ITO/Ag) is distributed on the two sides of a flexible Mo foil substrate. To keep bifacial consistency, the films of front side and back side are designed to realize simultaneously alternating depositions. The double-sided processes of substrate cleaning, spin-coating, annealing, selenization and CBD are respectively operated on one-time experiment. All the layers of the two sides are identical in thickness and crystallinity. The bifacial flexible CZTSSe solar cell obtained PCEs of 9.3% for front side and 9% for back side, which are the highest efficiencies for solution method based flexible CZTSSe solar cells. Meanwhile, the devices also show good performance of 6–8.8% efficiency under weak LED light illuminations with 1.5–18.5 mW/cm$^2$ intensities. The linear $J_{SC}$–$I_{LED}$ relationship reveals good weak light responses for the solar cells. In addition, the bifacial flexible CZTSSe devices standing the stretching and squashing stresses can maintain more than 95% of original efficiencies after bending 3000 times at the angle of 50°, exhibiting excellent flexibility. For the flexible performance of bifacial solar cells, numerous stretching

bending times causes grain boundary cracks and large angle squashing bending (>60°) on back side produces bottom peeling off, which are harmful to device performance. For parallel circuit of two sides, the output short-circuit current (8.7 mA) achieves the superposition of front (7.0 mA) and back cells (2.1 mA) with <5% loss, significantly improving the area utilization rate. The bifacial device could directly utilize all-directional lights and produce periodically variable currents by substrate rotation. The investigations of bifacial flexible CZTSSe solar cells provide a new prospect for indoor ornament integrated photovoltaics.

## Methods

**Double-sided CZTSSe film deposition.** The double-sided CZTSSe film was deposited by spin-coating and selenization annealing method. The precursor solution was obtained by dissolving the five powders of Cu (69.85 mg), Zn (49.7 mg), Sn (85.7 mg), S (86.4 mg) and Se (23.7 mg) in the mixed liquid of 1,2-ethanedithiol (0.5 mL) and ethylenediamine (5 mL), and stirring 2.5 h at 70 °C. The 1 mL stabilizer solution (HOCH$_2$CH$_2$NH$_2$, HSCH$_2$COOH, and HOCH$_2$CH$_2$OCH$_3$ as 1:1:2) was injected into the solution. One side of the Mo foil (marked as front side) was put on spin coater with a reticular diaphragm (see Fig. 2a). The front prefabricated layers were prepared by spin-coating the precursor solution on Mo foils at 3000 rpm for 30 s and baked on a hot plate (300 °C) for 1 min in N$_2$ glovebox. After cooling down to room temperature, the other side of Mo foil (marked as back side) was put on the spin coater with a clean reticular diaphragm. The prefabricated layers on the back side were obtained by the same spin-coating method. The process was repeated 9 times by alternating spin-coating on the front and back side to obtain prefabricated CZTSSe film (see Fig. 2a). The Mo substrate with prefabricated film was built on stilts near Se powder and put into a graphite box. The selenization annealing was conducted in a N$_2$ filled rapid annealing furnace (OTF-1200X) at 550 °C for 15 min to obtain double-sided crystalline CZTSSe films.

**Device fabrication.** Bifacial flexible CZTSSe solar cells were fabricated as the symmetrical device structure of Mo foil/MoSe$_2$/CZTSSe/CdS/ZnO/ITO/Ag. The Mo foils were divided into squares with 2 × 2 cm$^2$. The double sides of pieces were simultaneously chemically-polished in the sulfuric acid and methanol solution (H$_2$SO$_4$:CH$_3$OH = 7:1) by applying a voltage of 2.5 V for 2 min to remove the impurity and oxides. Double-sided CZTSSe film deposition and crystallization on clean Mo foil were fabricated by the above process. The double-sided CdS layers were simultaneously deposited on front and back CZTSSe films by chemical bath deposition (CBD) method. The CBD solution contained CdSO$_4$ (0.02 M, 20 mL), thiourea (0.68 M, 25 mL), ammonium hydroxide (20 mL), and deionized water (220 mL). The samples were exposed two sides and submerged in the CBD solution for 16 min at 70 °C. The obtained CdS film was annealed on a hot plate at 80 °C for 30 min. Double-sided ZnO and ITO layers were prepared by RF magnetron sputtering. Front and back sided ZnO layers with 60 nm thickness were successively deposited on CdS film for 8 min at the vacuum pressure of 2 Pa. Double-sided ITO layers with 200 nm thickness were successively deposited for 33 min at 0.5 Pa. Double-sided Ag electrodes with 400 nm thickness were fabricated twice by thermal evaporation under a vacuum pressure of 8 × 10$^{-4}$ Pa. Every device is divided by blade and its area is 0.21 cm$^2$ except grid area.

**Material characterizations and device measurements.** Surface and cross-section morphologies were observed by a scanning electron microscope (SEM, Helios G4 CX). The cross-section specimens were made using a focused ion beam (FIB, Helios G4 CX) and three ion beam cutting instrument (EM TIC 3X). The elemental distributions were measured by an energy dispersive spectroscopy (EDS). Crystal structure and orientation were investigated by an X-ray diffraction (XRD) spectra with Cu Kα radiation (Philips, x pert pro MRD). Current density–voltage ($J$–$V$) measurements for solar cells were illuminated by a Solar Simulator (SUN 2000, ABET) with a power density of 100 mW/cm$^2$. The samples were tested in air with the humidity of 60–80%. All the $J$–$V$ curves were obtained in forward direction from −0.5 V to 0.5 V. The statistical analysis was from 18 devices in each group. The external quantum efficiency (EQE) spectra were tested using a QTest Station500AD system equipped with a 150 W xenon light source, a lock-in amplifier, and an integrating sphere. Continuous illumination stability of solar cells was measured by a solar cell aging chamber (ATLAS SUNTEST CPSþ) with 1 sun light. For weak

light performance testing, the monochromatic light source for photoresponse testing was a 530 nm LED (Thorlabs M530L3) modulated by a waveform generator (Agilent 33600A Series). Light intensity was calibrated using a silicon photo-detector (Newport 818-UV). For flexible property testing, the bending angles were realized by the distance of two terminals assisted with a vernier caliper.

**Reporting summary**. Further information on research design is available in the Nature Research Reporting Summary linked to this article.

## Data availability

The data that support the findings of this study are available from the corresponding author (sycheng@fzu.edu.cn) upon reasonable request.

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

## Acknowledgements

This work was supported by the National Natural Science Foundation of China (No. 62074037, 61574038, and 52002073), the Natural Science Foundation of Fujian Province (2020J05105, 2020I0006), the Education and Scientific Research Project of Fujian Province (JAT190010), the Scientific Research Foundation of Fuzhou University (GXRC-20030). The authors also thank Testing Center of Fuzhou University for facility access.

## Author contributions

H. Deng and S. Cheng conceived the idea and designed the overall experiments, device fabrication, and measurement methods. Q. Sun, Z. Yang, and Q. Yan carried out the device fabrications and optimizations. H. Deng and W. Li contributed to characterizations, measurements, and analysis. Y. Lai, Q. Zheng, C. Zhang, and X. Wang and participated in the discussion and revision of the manuscript and data analysis. H. Deng mainly wrote and revised the manuscript. All of the authors discussed the results and commented on the manuscript.

## Competing interests

The authors declare no competing interests.
