## [Peer Review File · Nature Communications]

Reviewers' comments:

Reviewer #1 (Remarks to the Author):

Dear authors,

I am pleased to recommend the publication of your NCOMMS-20-47327 manuscript for this journal. The work you have shown here demonstrates the potential of bifacial flexible CZTSSe-based solar cells. I hope it will inspire scalable approaches for the fabrication of bifacial flexible kesterite solar cells. Before the publication of your results, please consider the following points.

1. The authors should rewrite this sentence in the introduction to make it clear: "Our group aim at developing low-cost solution methods for flexible CZTSSe solar cells on Mo foils and gradually improve efficiency to 7.19%."
2. Can the authors comment on the PCE after 1-hour illumination?
3. Apart from the introduction, the English writing needs a careful revision.

Reviewer #2 (Remarks to the Author):

This manuscript describes novel symmetrical bifacial CZTSSe solar cells. It is difficult to make high efficiency bifacial solar cells with transparent electrodes. In this manuscript, the authors tried to make bifacial solar cells by making double sided structure solar cells on both sides. The Mo foil was shared as an electrode of bifacial solar cell. I think it is pointless to use a conductive substrate as shared electrodes because the substrate and PV device must be separated when making PV modules.

This manuscript is need for revision.

- The authors wrote the most grain sizes of CZTSSe layers exceed 1 μ m. However, Fig3(c), (e) show there are many small grains in the bottom CZTSSe.
- The authors wrote the evolution of elements matched well with the stoichiometric proportion of every layer. However, Fig3(g) shows composition ratio and porosity are different depending on the vertical position. Please add the S component data in Fig3(g).
- The authors used the reflector which reflect about 0.46 times the standard sun light. The average albedo of the earth is 0.3. The light does not come in well to the back side because the light is obscured. Please explain why they used the reflector which reflect about 0.46.
- The I-V characteristics of the PV in lines 153 and 154 are different from those in Fig4(b). Please correct them.

Reviewer #3 (Remarks to the Author):

In this manuscript, the authors developed double side bifacial CZTSSe solar cells for indoor application. The concept of the CZTSSe device seems interesting, and the authors provide well organized research data and various characterizations in the manuscript. However, the authors do not provide comprehensive discussion for electrical properties of the device in the manuscript. Aslo, the results and discussion part are a simple listing of the results, and lacks scientific explanation, so it seems insufficient to attract readership. Unlike its title, the authors did not perform electrical characterization under indoor light condition. The device has different structure from conventional CZTSSe solar cells, we cannot assume the performance of the device under LED or Fluorescent lamp. Overall, I do not recommend it to be published at Nature Communication.

1. The title of the manuscript is double side bifacial CZTSSe solar cells for indoor application. However, in the experiments, the authors only conducted electrical characterizations close to 1 sun condition. The indoor light is much weaker (100~1000 times less) than 1 sun and the spectrum is also different, so I recommend to the authors to test the bifacial device under indoor light condition (LED or Fluorescent lamp).
2. Table 1. In general, FF is affected by Voc including ideality factor. The another side shows better FF even at lower light intensity. However, there is no discussion or explanation for this.
3. Line 206, The authors claims the loss of output power by bifacial is 14%. there is no detailed discussion or explanation for loss mechanism or causes.

Dear Editor and Reviewers,

Thank you very much for your comments and suggestions related to our manuscript! According to the suggestions and comments from the reviewers, we have revised all the corresponding points one by one. The detailed response is as following:

Reviewer: #1

Comment: I am pleased to recommend the publication of your NCOMMS-20-47327 manuscript for this journal. The work you have shown here demonstrates the potential of bifacial flexible CZTSSe-based solar cells. I hope it will inspire scalable approaches for the fabrication of bifacial flexible kesterite solar cells. Before the publication of your results, please consider the following points.

Reply: Thanks for your comments.

1. The authors should rewrite this sentence in the introduction to make it clear: “Our group aim at developing low-cost solution methods for flexible CZTSSe solar cells on Mo foils and gradually improve efficiency to 7.19%.”

Reply: Thanks for your suggestions. We have corrected the sentence in the revised manuscript.

Revised content: “Our group focuses on the study of Mo-foil-based flexible CZTSSe solar cells and has obtained 7.19% efficiency by technique optimization”. (Line 9-11, Page 3)

2. Can the authors comment on the PCE after 1-hour illumination?

Reply: Thanks for your suggestions. Stability of solar cell is an important index for practical application. We have checked the light stability of flexible CZTSSe solar cells by continuous illumination and added related data in Fig. S2b (Supporting Information). The device was put in a solar cell aging box by continuous illumination for several days and tested once a day. The unencapsulated devices have no obvious degradation (<5%) after 14-day continuous illumination. Flexible CZTSSe thin film solar cells using all inorganic materials reveal high stability which is hoped to realize wide application. We arranged these stability data to Supporting Information in the revised manuscript.

Revised content: “Stability of solar cell is an important index for practical application. The light

stability data of flexible CZTSSe solar cell is shown in Fig. S2a and S2b (Supporting Information). The PCE of the solar cells obtained a little increase after standing 5-min illumination and kept stable 9.3% for 1-hour illumination by continuous testing. Furthermore, the unencapsulated device was put in a solar cell aging box by continuous illumination with a standard sun intensity for several days and tested once a day. After 14-day continuous illumination, the device had no obvious degradation (<5%). Flexible CZTSSe thin film solar cells using all inorganic materials reveal high stability which is hoped to realize wide application.” (Line 1-8, Page 7)

Fig. S2. (a) PCE stability testing by continuous illumination for 1 hour. (b) PCE evolutions by continuous illumination for several days. The solar cell device was tested once a day.

3. Apart from the introduction, the English writing needs a careful revision.

Reply: Thanks for your suggestions. We have carefully checked English writing and corrected some mistakes in the revised manuscript.

Reviewer: #2

Comment: This manuscript describes novel symmetrical bifacial CZTSSe solar cells. It is difficult to make high efficiency bifacial solar cells with transparent electrodes. In this manuscript, the authors tried to make bifacial solar cells by making double sided structure solar cells on both sides. The Mo foil was shared as an electrode of bifacial solar cell. I think it is pointless to use a conductive substrate as shared electrodes because the substrate and PV device must be separated when making PV modules. This manuscript is need for revision.

Reply: Thanks for your comments. Bifacial solar cells have been widely studied which showed great potential for applications.¹⁻⁴ Bifacial Si solar cells have already been used in commercial at

bifacial modules which can collect albedo radiation and in static concentrator systems in which solar rays not only illuminate the front side of the cell but also are introduced to the rear side by mirrors. The bifacial silicon solar cell is with p-n junctions on two sides of a piece of Si. The presented bifacial flexible CZTSSe solar cells based on Mo foil substrate can realize same functions. The Mo foil with 60 μm thickness owns better strength and flexibility than Si substrate. The Mo foil used as a common substrate of bifacial solar cell reveals unique advantages in performance and costs. (1) Mo material is an important component of CZTSSe solar cells. Mo foil is directly used as the electrode and substrate, which reduces the additional process costs. (2) The front and back sides can independently work at opposite illumination and jointly output current by parallel circuit due to high conductivity of Mo foil. (3) The devices own good flexibility and high efficiency, providing an indoor application foundation. In this work, we mainly proposed the novel prototype device and studied the deposition techniques, measurements and device performances. We believed the Mo foils can be directly used as supported substrates in bifacial flexible CZTSSe devices and PV modules for indoor photovoltaic applications such as PV decorations. We will make further study for PV modules in future work.

References

1. Chu, Y.-W., et al. Fully Solution Processed, Stable, and Flexible Bifacial Polymer Solar Cells. *IEEE J. Photovoltaics* **10**, 508-513 (2020).
2. Ohtsuka, H., et al. Bifacial Silicon Solar Cells with 21.3% Front Efficiency and 19.8% Rear Efficiency. *Prog Photovoltaics* **8**, 385-390 (2000).
3. Fertig, F., et al. Economic feasibility of bifacial silicon solar cells. *Prog. Photovolt: Res. Appl.* **24**, 800-817 (2016).
4. Hezel, R. Novel applications of bifacial solar cells. *Prog. Photovolt: Res. Appl.* **11**, 549-556 (2003).

1. The authors wrote the most grain sizes of CZTSSe layers exceed 1 μm . However, Fig3(c), (e) show there are many small grains in the bottom CZTSSe.

Reply: Thanks for your comments. The description about grain sizes was from surface SEM image of CZTSSe film. The surface large-grain layer of CZTSSe film with $\sim 1 \mu\text{m}$ grains reveals high quality crystallization. The CZTSSe film fabricated by solution method generally emerges two layers containing 620 nm large-grain layer and 1500 nm small-grain layer, which is the optimized thickness of each layer in our previous work. We have revised the related expressions in the revised manuscript.

Revised content: “The CZTSSe films after selenization on surface of both two sides are smooth

and compact with clear grains (see in Fig. 3b and 3d). The surface large-grain layer of CZTSSe film with ~1 μm grains reveals high quality crystallization.” (Line 23-26, Page 5)

“The CZTSSe film deposited by solution method generally shows two layers, large-grain layer and small-grain layer, consistent with the related reports. The thicknesses of the two layers are 620 nm and 1500 nm, respectively.” (Line 4-6, Page 6)

2 The authors wrote the evolution of elements matched well with the stoichiometric proportion of every layer. However, Fig3(g) shows composition ratio and porosity are different depending on the vertical position. Please add the S component data in Fig3(g).

Reply: Thanks for your comments and suggestions. For the EDS measurement, the S and Mo elements have the same peak position. The testing system cannot distinguish the line scans of S and Mo elements (see Fig. R1). This testing issue also existed in literature reports (J. Mater. Chem. A, 2019, 7, 24891–24899). In fact, the Mo component from substrate far exceeds S component. The curve data completely conformed to the trend of Mo component trend which masked S signal. We inferred S component evolution tendency by Se and other elements. We have corrected the inaccurate expressions in the revised manuscript.

Fig. R1. The tested Mo and S component data.

Revised content: “The evolutions of all elements are essentially consistent with the materials of every layer. It can be observed the elements of Cu, Zn, Sn, Se distributed in absorber layer. The S element signal is masked by stronger signal of Mo element. From surface to bottom of CZTSSe film, the contents of metal cations (Cu, Zn, Sn) reduce and those of anions (Se, S) increase, which mainly cause the small-grain layer. The average composition of absorber is $\text{Cu}_2\text{ZnSn}(\text{S}_{0.17}\text{Se}_{0.83})_4$. The junction interface and back contact exist element interdiffusion, which enhance the anchoring strength with substrate.” (Line 8-14, Page 6)

3. The authors used the reflector which reflect about 0.46 times the standard sun light. The average albedo of the earth is 0.3. The light does not come in well to the back side because the light is obscured. Please explain why they used the reflector which reflect about 0.46.

Reply: Thanks for your comments and suggestions. We used the 0.46 times the standard sun light because the reflector can reach the maximum reflection of sun light to another sides. The value had no special significance. As the average albedo of the earth is 0.3, we changed the reflection light intensity to 0.3 times the standard sun light and tested the performance (seen in Fig. 6 and Table 1). In addition, we added the weak-light performance data of CZTSSe solar cells under LED light illumination as shown in Fig. 4d-f and S3.

Revised content: “Considering that average albedo of the earth is 0.3, We use a reflector to change the light direction and reflect about 0.3 times the standard sun light to another sides. The output current-voltage (I-V) curves for the bifacial solar cells are shown in Fig. 6b. The solar cell of one side illuminated at 1 standard sun obtained 9.1% efficiency. And the another sided cell at the 0.3 times sun illumination obtained 8.7% efficiency with decreased V_{OC} (seen in Table 1).”

(Line 4-8, Page 9)

“For the parallel solar cell, the V_{OC} (416 mV) is between those of front (432 mV) and back cells (406 mV). The output short-circuit current (I_{SC}) of parallel device (8.7 mA) is roughly equal to the sum of independent currents of front cell (7.0 mA) and back cell (2.1 mA). The loss rate of output current was less than 5% in the parallel circuit. From the power curves in Fig. S6 (Supporting Information), the maximum output power (2.05 mW) was also approximate to the sum of the two sides with about 15% loss.” **(Line 14-20, Page 9)**

Table 1. The typical parameters of double sided CZTSSe solar cells in parallel.

	Light	V_{oc} (mV)	I_{sc} (mA)	J_{sc} (mA/cm ²)	FF	PCE	P_{max} (mW)
One side	1 Sun	432	7.0	33.4	63%	9.1%	1.87
Another side	0.3 Sun	406	2.1	10.1	64%	8.7%	0.55
Parallel	/	416	8.7	/	58%	/	2.05

4 The I-V characteristics of the PV in lines 153 and 154 are different from those in Fig4(b). Please

correct them.

Reply: Thanks for your reminding. We have corrected the mistakes in the revised manuscript.

Revised content: “The optimal efficiency of front cell is 9.3% with a V_{OC} of 436 mV, a J_{SC} of 33.76 mA/cm² and an FF of 63.2%. Similarly, the back cell obtains 9% PCE with the V_{OC} of 434 mV, the J_{SC} of 33.7 mA/cm² and the FF of 61.7% (seen in Fig. 4b).” (Line 26-28, Page 6)

Reviewer: #3

Comment: In this manuscript, the authors developed double side bifacial CZTSSe solar cells for indoor application. The concept of the CZTSSe device seems interesting, and the authors provide well organized research data and various characterizations in the manuscript. However, the authors do not provide comprehensive discussion for electrical properties of the device in the manuscript. Also, the results and discussion part are a simple listing of the results, and lacks scientific explanation, so it seems insufficient to attract readership. Unlike its title, the authors did not perform electrical characterization under indoor light condition. The device has different structure from conventional CZTSSe solar cells, we cannot assume the performance of the device under LED or Fluorescent lamp. Overall, I do not recommend it to be published at Nature Communication.

Reply: Thanks for your comments and suggestions. In this work, we proposed a novel symmetrical bifacial flexible CZTSSe solar cells for indoor ornament integrated photovoltaic applications. We systematically studied fabrication techniques, material characterization, bending properties and parallel circuit performance. The study is expected to provide innovative fundamental devices and designs to expand photovoltaic applications. According to the suggestions, in the revised manuscript, we have added the performance characterization of CZTSSe solar cells under indoor light condition by LED illumination as shown in Fig. 4d-f and S3. We have also supplemented discussion and explanation for electrical properties.

1. The title of the manuscript is double side bifacial CZTSSe solar cells for indoor application. However, in the experiments, the authors only conducted electrical characterizations close to 1 sun condition. The indoor light is much weaker (100~1000 times less) than 1 sun and the spectrum is also different, so I recommend to the authors to test the bifacial device under indoor light condition (LED or Fluorescent lamp).

Reply: Thanks for your comments and suggestions. We have supplemented the performance characterizations of CZTSSe solar cells under LED light illumination as shown in Fig. 4d-f and S3 in the revised manuscript.

Revised content: “To check the indoor weak light response, the performance of flexible CZTSSe solar cells were tested using a LED light illumination. The J-V characteristics of devices under the LED light intensities of 1.5-18.5 mW/cm² are shown in Fig. 4d. All the curves with rectification effects reveal good response for weak light. The device owns 8.8% efficiency when the light intensity is 18.5 mW/cm². And the PCE can maintain 6% even illuminated at 1.5 mW/cm² weak LED light. The PCE, J_{SC}, V_{OC}, and FF evolutions with light intensities are shown in Fig. 4e, f and Fig. S3a, b. The J_{SC} values show superb linear relationship with LED light intensities (I_{LED}) (see in Fig. 4e). We fitted J_{SC}-I_{LED} relationship according to the power law dependence (J_{SC} ∝ I^α) by logarithmic coordinates. The power value α for the devices is 0.99 in weak light condition, close to unity (first-order). This means that the CZTSSe solar cell demonstrates good photoresponse for weak light and its loss mechanism is from trap-assisted recombination. The recombination process can be reflected based on the relationship of V_{OC} ∝ n(k_BT/q)ln(I), where k_B is the Boltzmann constant, T is the temperature and q is elementary charge. For trap-free solar cells, the slope of V_{OC} vs. ln(I) should be k_BT/q (i.e., n = 1). The V_{OC} under weak light illumination evolutions with logarithms of light intensities accord with the relationship V_{OC} = 1.42(k_BT/q)ln(I_{LED}) as shown in Fig. S3a (Supporting Information). The larger slope indicated V_{OC} would rapidly decline because of trap recombination under the extremely weak light. The decreased photogenerated carriers at weak light condition weaken V_{OC} and J_{SC} of solar cells. While the fill factors can keep constant at ~64% for LED illumination with 4-18.5 mW/cm² intensities (Fig. S3b, Supporting Information). the FF is mainly affected by interface and contact of each layer without change by illumination. When the light intensity is less than 4 mW/cm², the fill factors show obvious decrease. From the efficiency trend in Fig 4f, the PCEs of flexible CZTSSe solar cells are still more than 8% under the LED light with 5-20 mW/cm² intensities. The performances manifest degradation at extremely weak light (<3 mW/cm²) mainly caused by decreased V_{OC} and FF. On the whole, the presented bifacial CZTSSe flexible solar cells can realize weak light response and be suitable for indoor photovoltaic application.” (Line 9-33, Page 7)

Fig. 4. (a) Statistical PCE distribution boxplots of front and back sides on bifacial flexible CZTSSe solar cells. (b, c) The J-V curves (b) and EQE spectra (c) of champion double sided cells. (d) The J-V curves of front solar cell under the LED light illuminations with different intensities. (e) The obtained J_{sc} evolution with LED light intensity. (f) The obtained PCE evolution with LED light intensity.

Fig. S3. (a, b) The obtained V_{oc} (a) and Fill factor (b) evolutions with LED light intensity.

2. Table 1. In general, FF is affected by V_{oc} including ideality factor. The another side shows better FF even at lower light intensity. However, there is no discussion or explanation for this.

Reply: Thanks for your comments and suggestions. According to the above study for weak light performance, J_{sc} decreases as linear ($J_{sc} \propto I_{LED}$) trend and V_{oc} decreases as logarithmic trend ($V_{oc} \propto 1.42(k_B T/q) \ln(I_{LED})$) with the decreased light intensity. For the fill factor, it can keep constant at $\sim 64\%$ for LED illumination with 4-18.5 mW/cm² intensities (Fig. S3b, Supporting Information). The fill factors of back side solar cell at 0.46 times or 0.3 times sun also showed 64-65%, which are a little higher than that of the devices at 1 sun. On the one hand, the FF is

mainly affected by interface and contact of each layer without change by illumination. The less photogenerated carriers at weak light also decrease the recombination loss of interfaces and improve FF. On the other hand, the temperature on sample at weak light condition is lower than that of 1 sun illumination, which is helpful to suppress trap recombination and improve reverse saturation current (J_0) for higher FF. However, when the light intensity is extremely weak (less than 4 mW/cm^2) (Fig. S3b), the fill factors dominated trap recombination in solar cells show obvious decrease, indicating same trend with V_{OC} . We have supplemented discussion and explanation in the revised manuscript.

Revised content: “The photogenerated carriers come to decrease at weak light condition, the V_{OC} is decreased as logarithmic trend and J_{SC} decreased as linear trend with light intensity according to the above analysis. The FF of solar cell (64%) at 0.3 times sun is a little higher than that of device (63%) at one sun light. The temperature on sample at weak light condition is lower than that of 1 sun illumination, which is helpful to suppress trap recombination and improve reverse saturation current for higher FF.” (Line 9-14, Page 9)

3. Line 206, The authors claim the loss of output power by bifacial is 14%. there is no detailed discussion or explanation for loss mechanism or causes.

Reply: Thanks for your comments and suggestions. The front and back cells produce different V_{OC} (432 mV and 406 mV, respectively) under different illuminations. And the parallel device obtains the V_{OC} of 416 mV between those of front (432 mV) and back cells (406 mV). For the parallel circuit of double sides, the front cell with higher V_{OC} provides some voltage consumption for the back cell. Therefore, the parallel power loss is from the internal resistance consumption resulted by unequal doubled V_{OC} . We have supplemented related explanation in the revised manuscript.

Revised content: “From the power curves in Fig. S6 (Supporting Information), the maximum output power (2.05 mW) is also approximate to the sum of two (1.87 and 0.55 mW) with about 15% loss. The front and back cells produce different V_{OC} (432 mV and 406 mV, respectively) under different illuminations. In the parallel circuit, the front cell provides some voltage consumption for the back cell. Therefore, the parallel power loss is from the internal resistance consumption resulted by unequal doubled V_{OC} , which is acceptable for largely different intensities of two sides. The bifacial CZTSSe solar cells have revealed parallel superimposed effect for current and output power even illuminated at different illumination conditions of two sides.” (Line 18-25, Page 9)

REVIEWERS' COMMENTS

Reviewer #1 (Remarks to the Author):

Dear Authors,

I am pleased to recommend the publication of your NCOMMS-20-47327 manuscript for this journal after the corrections you have made.

Reviewer #2 (Remarks to the Author):

This manuscript is suitable for publication in Nature Communication due to its appropriate modifications.